# Inter-Rater Reliability of the Structured Interview of DSM-IV Personality (SIDP-IV) in an Adolescent Outpatient Population

**DOI:** 10.3390/ijerph191912283

**Published:** 2022-09-27

**Authors:** Eirik Wixøe Svela, Hans Ole Korsgaard, Line Indrevoll Stänicke, Randi Ulberg

**Affiliations:** 1Psychiatric Outpatient Clinic, Lovisenberg Hospital, 0653 Oslo, Norway; 2The Nic Waal Institute, Lovisenberg Hospital, 0853 Oslo, Norway; 3Department of Psychology, University of Oslo, 0315 Oslo, Norway; 4Division of Mental Health and Addiction, University of Oslo, 0315 Oslo, Norway; 5Department of Psychiatry, Diakonhjemmet Hospital, 0373 Oslo, Norway; 6Division of Mental Health, Vestfold Hospital Trust, 3125 Tønsberg, Norway

**Keywords:** adolescence, personality disorder, psychometric testing

## Abstract

Personality disorders (PDs) are a class of mental disorders which are associated with subjective distress, decreased quality of life and broad functional impairment. The presence of one or several PDs may also complicate the course and treatment of symptom disorders such as anxiety and depression. Accurate and reliable means of diagnosing personality disorders are thus crucial to ensuring efficient treatment planning and resource allocation, a fact which is widely acknowledged within the adult mental health field. In adolescents, on the other hand, the consensus view has been that the rapid and discontinuous processes of normal personality development render the construct of PD in adolescents clinically unhelpful and conceptually dubious. However, recent research has established the reliability and validity of the construct, heralding an increased interest in the field, with important conceptual and clinical implications. The present article presents reliability data for the Structured Interview of DSM-IV Personality (SIDP-IV) in an adolescent outpatient population. For this study, 25 interviews conducted by experienced clinicians were blindly re-scored based on sound recordings, which allowed for the calculation of intra-class correlation metrics. The intra-class correlation coefficient for categorical diagnosis of specific personality disorders was found to be 0.876 (95% CI 0.718–0.945); Cohen’s kappa for presence/absence of personality diagnosis was found to be 0.896. The present study found excellent intra-rater reliability for the sample, which suggests that the SIDP-IV is a suitable instrument for assessing personality pathology in adolescent populations.

## 1. Introduction

Personality disorders (PDs) are a group of mental disorders characterized by global disturbance of relationships, self-conception, affect, and cognition [1]. Personality disorders are associated with significant morbidity [2], functional impairment [3], increased risk of suicide [4], as well as increased health service use [5], and decreased quality of life [6,7,8]. The lifetime prevalence of all personality disorders among adults is estimated to be 7.8% [9], while the cumulative lifetime prevalence has been estimated at 28% at the age of 33 [10].

While the DSM-5 stipulates specific, individual personality disorders, most researchers today acknowledge the existence of widespread comorbidity between individual personality disorders, significant comorbidity between personality disorders and symptom disorders, and significant manifestations of dysfunctional personality traits in individuals who do not meet the threshold of any one personality disorder diagnosis [11]. Consequently, many writers have adopted a dimensional model of personality pathology in which the specific personality disorders may be only the most fulminant manifestations; indeed, this reasoning appears to be implicit in many theoretical models of personality pathology, such as Kernberg’s work [12,13] and the schema model [14]. Furthermore, this development is reflected in the adoption of a dimensional system of classification in the ICD-11, as well as the DSM-5 alternative model for personality disorders [15], both of which aim to acknowledge and integrate these insights. For the purposes of this article, *personality disorder* will be used to refer to concrete personality disorders, whereas *personality pathology* will refer to this broader usage. 

Albeit traditionally considered difficult to treat, the last five decades have seen the emergence of an array of treatment approaches that grapple with personality pathology from differing positions on (i) the essential properties of human personality and its development [16,17], (ii) the putative pathogenic processes involved in personality pathology [18], (iii) continuity or discontinuity between “normal” personality and personality pathology [19], (iv) the diachronic and situational stability of normal personality and personality pathology [20], and (v) the relative etiological significance of biogenetic, shared environmental, and non-shared environmental forces on personality development and pathology [21]. Several treatment approaches are currently in widespread use [22,23]. Transference-focused psychotherapy [24] and mentalization-based therapy [25] are examples of treatment approaches that proceed from the assumptions of psychoanalytic clinical thought, whereas dialectical-behavioral therapy [26] and schema therapy [14] are approaches emerging from the cognitive-behavioral tradition.

Most approaches to personality pathology are informed by theories of normal personality, which implicitly or explicitly assume a developmental trajectory in which an individual’s personality may not be considered fully mature and stable until the end of adolescence, or perhaps later still [11,27,28]. Consequently, the consensus view has been that the rapid and discontinuous processes of normal personality development render the idea of personality disorders in adolescents untenable. Furthermore, many clinicians have held that early PD diagnosis could lead to unnecessary stigma for the young person in question, or that such a diagnosis might contribute to the chronification of a symptomatic pattern which might otherwise be responsive to treatment. 

Over the last two decades, however, researchers have begun to question this assumption. Chanen and colleagues made an important early contribution by demonstrating that DSM-IV personality disorder categories show the same reliability over two years as personality disorders in adults [29]. A growing body of studies has extended these findings with data on validity [30], prevalence [20], and comorbidity [31]. The validity of personality disorder categories in adolescent populations thus appears to be a question of theoretical as well as clinical significance. 

The effectiveness of specific treatment programs aimed at adolescent patients with personality pathology is also becoming an object of study [11,32,33]. These efforts have yet to produce evidence for the overall effectiveness and superiority of any one particular treatment, raising important questions about the specific manifestations of personality pathology in this population, appropriate research designs, and the nature of personality development, the answers to which could prove crucial in tailoring treatment strategies for this population [34].

A separate question is whether early assessment and diagnosis of personality disorders—and consequent implementation of treatment—can prevent or mitigate personality-related morbidity and/or comorbid symptom pathology [35]. Regardless of treatment outcome, early detection of significant personality pathology may prove meaningful and useful to the patient. These research endeavors rest on the assumption that PDs may be reliably and validly diagnosed in adolescents. Effective, efficient, and reliable means of assessment are an important prerequisite for this. However, existing studies have mostly proceeded on the assumption that existing instruments and methods of assessment are appropriate for adolescent populations with few and minor alterations; few studies have specifically examined the psychometric properties of these instruments in adolescent populations. Psychometrics may be defined as “the branch of psychology concerned with the quantification and measurement of mental attributes, behavior, performance, and the like, as well as with the design, analysis, and improvement of the tests, questionnaires, and other instruments used in such measurement” [36]. Psychometric analysis allows researchers to model and study latent mental phenomena which are not directly observable, such as personality traits or intelligence. The dearth of psychometric validation arguably undermines the confidence of the conclusions that may be gleaned from existing studies.

As of this writing, only the United Kingdom [37,38] and Australia [39] have elaborated national guidelines for the assessment and diagnosis of personality disorders in adolescents. However, such routines and guidelines are currently under preparation in several countries [40]. In our opinion, research establishing the psychometric properties of commonly used instruments of assessment and diagnosis may aid decision-makers and other stakeholders in determining which tools and instruments to recommend for routine usage.

The instrument examined in this article is the Structured Interview of DSM-IV Personality Disorders (SIDP-IV), which is a semi-structured diagnostic interview for diagnosing DSM-IV personality disorders [41]. Because the DSM-IV classification of personality disorders has been retained in largely unchanged form in the DSM-5, this psychometric instrument remains relevant. Unlike similar instruments, such as the Structured Clinical Interview for DSM-IV-Axis-II (SCID-II) [42], the SIDP-IV includes the “provisional” diagnoses listed in Appendix B of the DSM-IV, enabling the clinician to assess a broader variety of personality pathology. Furthermore, unlike similar instruments, items in the SIDP-IV are ordered thematically rather than by diagnostic category. This arguably facilitates a more fluent, conversational interview experience that may be particularly important in securing forthcoming and candid participation from adolescent subjects. Importantly, the SIDP-IV has seen extended application in clinical work with adolescents, for which it has been deemed robust and efficient [43]. The SIDP-IV also encourages routine solicitation of information from family members or close friends, which may be uniquely important in assessing personality pathology in adolescents. Finally, the SIDP-IV yields an array of diagnostic and subdiagnostic data which can easily be reanalyzed and utilized in a dimensional model, such as the DSM-5-AMPD [44] and ICD-11 [45] models.

The aim of the present study is to determine whether the SIDP-IV shows acceptable psychometric properties, specifically in terms of inter-rater reliability, in an adolescent sample.

## 2. Materials and Methods

### 2.1. Properties of the Sample

The data material under examination in this study was collected from the First Experimental Study of Transference—In Teenagers (FEST-IT) study [46]. The FEST-IT study is a randomized, double-blind dismantling study that aims to isolate and investigate specific effects of psychodynamic transference interpretations in psychotherapy with adolescents. The rationale for transference work is an acknowledgment that “the ongoing interaction between patient and psychotherapist is heavily influenced by the patient’s past relationships and affective experiences”, and consists of “a focus on themes and conflicts that arise in the therapeutic relationship”, which is thought to “have immediate affective resonance and illuminate the nature of problems in the patient’s relationships outside therapy” [46]. Transference work is regarded as one of the defining components of psychodynamic psychotherapy. 

The study included 69 participants between the ages of 16 and 18, with a mean age of 17.3 years [47]. The primary inclusion criterion was Major Depressive Disorder, while exclusion criteria were substance addiction, psychosis, generalized learning difficulties, and pervasive developmental disorder. The majority of the participants were female (82.6%; *n* = 57). Symptom severity at pre-treatment was assessed using the Montgomery-Åsberg Depression Rating Scale [48] and the Beck Depression Inventory [49], yielding averages of 22.72 (MADRS) and 28.62 (BDI), suggesting moderate depression according to common scoring conventions. Symptom severity at post-treatment was assessed using MADRS, producing an average score of 9.5, suggesting only minor residual depressive symptoms. The average total SIDP-IV symptom score at pre-treatment was 13.02. 

The participants were randomized into two groups. Patients in the active control group received short-term psychodynamic psychotherapy, whereas patients in the study group received short-term psychodynamic psychotherapy; however, in this group, therapists were instructed to specifically refrain from using transference interventions.

Because we expected to find a good degree of inter-rater reliability, a relatively small sample size of 25 was deemed sufficient. Expecting to find a kappa coefficient of at least 0.6, a sample size of 25 is sufficient at 90% statistical power [50]. Anticipating an ICC coefficient of at least 0.7, a sample size as small as 13 is sufficient at 90% power [51]. The participants were included by selecting every third participant among the sequential subject IDs, thus minimizing systematic variance and ensuring the overall representativeness of the data. Nine participants were interviewed pre-treatment, nine participants were interviewed at post-treatment, and seven participants were interviewed for the follow-up study one year later. There were no systematic differences among these participants other than their participation status (pre-treatment, post-treatment, one-year follow-up). All participants in the sample (*n* = 25) were female, owing to the random effects of the sampling procedure.

### 2.2. The Structured Interview of DSM-IV Personality (SIDP-IV)

The SIDP-IV interviews were conducted and scored by a group of experienced clinical psychologists and psychiatrists who were blind to the randomization status of the patients, and audio recordings were taken of these interviews. The interviewers attended joint training sessions to ensure optimal adherence to the study protocols and standards. Based on these recordings, the primary author independently scored the interviews, blind to the initial scores and the randomization status of the patients.

The SIDP-IV consists of 101 questions and can generally be carried out within 60 to 90 min. The response to each item is assigned a score between 0 and 3. A score of 0 corresponds to no pathology. A score of 1 corresponds to some subthreshold pathology, operationalized as “some evidence of the trait, but it is not sufficiently pervasive or severe to consider the criterion present”. A score of 2 corresponds to pathology that is present most of the time, operationalized as “clearly present for most of the last 5 years (i.e., present at least 50% of the time during the last 5 years)”. A score 3 corresponds to marked manifestations of pathology, operationalized as “strongly present criterion is associated with subjective distress or some impairment in social or occupational functioning or intimate relationships” [41]. 

### 2.3. Properties of the Data Obtained

Scores were obtained at several levels of description/detail. First, the continuous total score of items endorsed at any level (0–3) was registered for each patient, an interval variable capturing the presence of minimal/subthreshold personality dysfunction as well as more pronounced pathology. This value can be useful in terms of detecting and assessing subclinical degrees of personality pathology, which could still be significant, especially when dispersed among diagnostic categories.

Second, criterion scores, meaning scores of 2 or 3, were registered. This interval value captures the presence of significant personality dysfunction, irrespective of individual personality disorder diagnosis. This value is useful for detecting and assessing significant pathology dispersed among categories, e.g., when patients do not satisfy the requirements for any one diagnosis but nevertheless exhibit clinically significant personality pathology.

Third, the number of individual personality disorders assessed as present, a hybrid ordinal-interval variable, was computed. This value provides information on the overall morbidity of an individual and sheds light on patterns of comorbidity among personality disorders. 

Finally, an ordinal/dichotomous variable captures the overall presence or absence of any PD diagnosis. This value serves as a useful proxy for the general personality pathology assessed for an individual, irrespective of the degree or character of pathology encountered.

### 2.4. Analyses

Inter-rater reliability metrics for the data material were computed. Two types of metrics were obtained: The intra-class correlation coefficient (ICC), a metric that is commonly used to compute correlation coefficients of interval/continuous variables, and Cohen’s kappa (*κ*), a metric that is commonly used to compute correlation coefficients of categorical/ordinal variables [52,53].

ICC coefficients were computed for (i) number of diagnoses assessed, (ii) criterion scores assessed, and (iii) continuous item scores. Cohen’s kappa was computed for (i) number of diagnoses assessed and (ii) presence or absence of any diagnosis.

For our purposes, the ratings conducted by the experienced raters were treated as though they were conducted by one person, on the reasoning that this greatly simplifies the analysis and because no meaningful divergences were expected in the group of experienced clinicians. A two-way, random design ICC model, referred to as ICC (2,2), was selected [54]. IBM SPSS allows for two modes of computing ICC (2,2), emphasizing either degree of consistency or absolute agreement. Because our data material primarily consisted of continuous data, the former algorithm was selected.

The formula for calculating the ICC coefficient is
$$\mathrm{ICC}=\frac{MS_{R}-MS_{E}}{MS_{R}}$$

The formula for calculating Cohen’s kappa is given below
$$\kappa \equiv \frac{p_{0}-p_{e}}{1-p_{e}}=1-\frac{1-p_{0}}{1-p_{e}}$$

IBM SPSS version 26 (IBM, Armonk, NY, USA) was used for the calculations [55].

### 2.5. Ethics

The consent of all participants was obtained in writing for their overall participation in the study, and because no further data collection was conducted, no further consent was deemed necessary. The study was approved by the Central Norway Regional Ethics Health Committee. FEST-IT. FEST-IT is registered in ClinicalTrials.gov (accessed on 18 September 2022): NCT01531101.

## 3. Results

### 3.1. Intra-Class Correlation Coefficients

The intra-class correlation coefficients for the sample (*n* = 25) are presented in Table 1. 

### 3.2. Cohen’s Kappa

The values for Cohen’s kappa are presented in Table 2 below.

## 4. Discussion

### 4.1. Summary of Findings

The values obtained for the intraclass coefficient and Cohen’s kappa were statistically significant and suggest high reliability for the SIDP-IV in this sample. This holds true not only for categorical diagnosis (i.e., reliability of individual PDs), but also for criterion scores (i.e., reliability of dimensional assessments of severity of personality severity) and continuous scores (i.e., reliability of sum of criterion scores).

### 4.2. Interpretation

According to one common convention for interpretation of ICC coefficients, ICC coefficients between 0.5 and 0.75 indicate moderate reliability, coefficients between 0.75 and 0.9 indicate good reliability, and coefficients greater than 0.9 indicate excellent reliability. Furthermore, it is generally recommended that ICC coefficient findings be reported in terms of 95% confidence intervals, as the true value of the coefficient could theoretically be on any point in this range [54]. 

The ICC coefficient for categorical diagnosis—i.e., for individual diagnoses, such as borderline personality disorder or paranoid personality disorder—was found to be 0.876, suggesting good reliability. The 95% confidence interval ranged from 0.718 to 0.945, corresponding to a range between moderate and excellent reliability. The ICC coefficient for criterion scores—i.e., endorsed items suggesting clear personality pathology—was found to be 0.952, suggesting excellent reliability for this metric. The 95% confidence interval ranged from 0.891 to 0.979, corresponding to a range of good to excellent reliability.

A common convention for interpreting Cohen’s kappa holds that values between 0 and 0.2 indicate poor agreement, values between 0.21 and 0.4 indicate fair agreement, values between 0.41 and 0.6 indicate moderate agreement, values between 0.61 and 0.8 indicate good agreement, and values between 0.81 and 1 indicate very good agreement [56]. Based on these criteria, the kappa coefficient for categorical diagnosis—i.e., presence and number of discrete diagnoses—was 0.623, indicating good agreement. The kappa coefficient for dichotomous diagnosis—i.e., overall presence or absence of a personality disorder diagnosis—was found to be 0.896, indicating very good reliability for this metric. All results were statistically significant at *p* < 0.0005, which indicates an exceedingly low likelihood that the null hypothesis was rejected in error. 

Thus, the results suggest that this instrument, originally designed for use with adult populations, retains a high degree of reliability when used with a sample of adolescent outpatients. 

### 4.3. Strengths and Limitations

The primary strength of the study is the fact that it is the first to examine the psychometric properties of this instrument in an adolescent population. In fact, to our knowledge, it is the first study of any PD diagnostic instrument to examine such a population, and so provides important initial confirmation of the notion that personality disorders may be reliably diagnosed in adolescents. Psychometric validation is crucial in ensuring that psychological tests and instruments are in fact as valid and reliable as they appear to be in clinical or research settings.

The main limitation of the study is the fact that it only examines female subjects, and as such, it is unclear how well the findings generalize across sexes. As far as we are aware, there is no information that would specifically lead us to expect different results in an all-male or mixed-sex population, but it seems that this question requires further study to be determined with certainty.

A further limitation similarly derives from the composition of the population under study. The patients who participated in the study were under treatment for moderate depressive disorders, and, perhaps predictably, most of the patients who met the criteria for PDs endorsed neurotic, anxious, and dependent/avoidant pathology, often referred to as Cluster C PDs. None of the participants included in this study endorsed items indicative of paranoid, schizoid, or schizotypal (Cluster A) PDs, and only some participants endorsed items indicative of pathology associated with emotional and/or behavioral instability, suggestive of a Cluster B PD. For this reason, the reliability figures obtained for Cluster A and Cluster B PDs suggest excellent reliability in terms of the absence of such pathology, but our data material does not speak to the reliability of the instrument in patients who do exhibit a high degree of such traits. Further research drawing on a more diverse population seems required to resolve this question.

We have opted to discuss the reliability of the SIDP-IV in terms of its inter-rater reliability. This focus springs from the data material available, which naturally lends itself to conducting an inter-rater reliability analysis, a metric which in our view is crucially important to the overall clinical utility and interpretability of a psychometric instrument. We encourage future research that will hopefully shed light on other aspects of the instrument’s reliability, such as the test–retest reliability of the SIDP-IV.

Finally, our analyses were conducted on a relatively small sample (*n* = 25). However, the strength of the findings—specifically, the high significance—suggests that it is highly likely that a study conducted with a larger population would produce similar psychometric characteristics. To illustrate this point, reliability metrics were computed with a stricter 99% confidence interval and are presented in Table 3 below. Indeed, this analysis yields broadly similar figures, none of which shows less than moderate reliability (i.e., lower range of interval for categorical diagnosis).

### 4.4. Implications and Future Directions

The results suggest that the instrument may be used with confidence with adolescent populations, at least insofar as reliability is concerned. Indeed, our findings were comparable to the results obtained for an adult sample [57], supporting the contention that personality diagnosis in adolescence is as reliable as in adulthood. 

However, the results have no bearing on the validity of the diagnostic information derived from the application of the SIDP-IV, which our data does not permit us to examine. More research is required to shed light on the convergent, divergent, external, and ecological validity of personality diagnosis based on the SIDP-IV, as well as for other instruments. As results accrue in support of the reliability of the SIDP-IV and similar instruments, validity data may provide conclusive evidence for the overall construct and notion of PD in adolescents. 

In the light of the move towards a dimensional classification of PDs, as heralded by the adoption of this model in the ICD-11 [58], the reader may question the relevance of novel work on instruments and procedures which posit individual diagnostic categories. However, the DSM-IV/ICD-10 system of classification has been retained in the DSM-5, which remains widely used in the United States, and is greatly influential in terms of future research, meaning that questions of the validity and reliability of these instruments remain highly relevant.

Furthermore, we would argue that the SIDP-IV remains valuable even when adopting a dimensional approach, as this instrument yields an abundance of data beyond the simple question of the presence or absence of individual diagnoses. While the SIDP-IV may be used to diagnose individual PDs, it also provides highly relevant information on several items endorsed suggesting pathology, number of items endorsed suggesting subclinical levels of personality dysfunction, as well as data useful in determining more nuanced profiles of personality dysfunction when no one diagnosis—or multiple diagnoses—may confidently be reached. Indeed, as seen in Table 1, the reliability of sub-diagnostic data—criterion and total scores—appears to be even higher than for diagnostic information. This suggests that data yielded by the SIDP-IV may be used with little or no modification within the framework of a dimensional model of personality pathology which may not take individual PDs as its starting point, but rather a broader, more inclusive conception of personality pathology. Such a development might perhaps also more readily accommodate the widespread patterns of comorbidity observed between PDs, and between PDs and symptom disorders, possibly suggesting underlying latent etiological factors held in common. From a clinical point of view, the co-occurrence of sub-diagnostic personality issues, e.g., as measured by the SIDP-IV, and symptom disorders in an adolescent patient may prove to be an apt starting point for early intervention. Hopefully, future research will determine whether such strategies carry the potential to mitigate the impact of personality issues, or possibly prevent an individual from developing a manifest personality disorder. It also suggests that the items contained in the SIDP-IV are excellent candidates for inclusion in novel instruments for assessing PD.

## 5. Conclusions

Our findings indicate a high degree of inter-rater reliability between the scores obtained by the primary author and those obtained by expert clinicians. An ICC coefficient of 0.876 was found for individual diagnoses and Cohen’s kappa was found to be 0.896 for dichotomous diagnosis, indicating good reliability for the SIDP-IV in this population. 

This suggests that the instrument may be used with confidence with adolescent populations and provides initial confirmation that diagnostic instruments that were designed and normed on adult populations are likely to prove reliable when adapted to adolescent populations. In preparing guidelines for routine assessment of PDs in child and adolescent mental health services, our data suggest that the SIDP-IV may be a promising candidate.

## Figures and Tables

**Table 1 ijerph-19-12283-t001:** Intra-class correlation coefficients with results of F-test.

	Individual Categorical Diagnoses	Criterion Scores	Continuous Scores
ICC	0.876	0.952	0.917
95% CI	0.718–0.945	0.891–0.979	0.808–0.964
Value of F-test	8.036	20.902	12.062
Degrees of freedom (*df*1 and *df*2)	24 and 24	24 and 24	23 and 23
F-test significance level	*p* < 0.0005	*p* < 0.0005	*p* < 0.0005

**Table 2 ijerph-19-12283-t002:** Kappa coefficients for categorical diagnosis and dichotomous diagnosis.

	Kappa Coefficient	Asymptotic Standard Error	*t*-Test Statistic	Significance
Categorical diagnosis	0.623	0.134	4.376	*p* < 0.0005
Dichotomous diagnosis	0.896	0.101	4.506	*p* < 0.0005

**Table 3 ijerph-19-12283-t003:** Intra-class correlation coefficients computed for 99% CI.

	Individual Categorical Diagnoses	Criterion Scores	Continuous Scores
99% CI	0.631–0.958	0.858–0.984	0.748–0.973
F-test significance level	*p* < 0.0005	*p* < 0.0005	*p* < 0.0005

## Data Availability

The data is available from the second author.

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
