# Peer review of "Inter-Rater Reliability of the Structured Interview of DSM-IV Personality (SIDP-IV) in an Adolescent Outpatient Population"

_ijerph, 2022, doi:10.3390/ijerph191912283_

Round 1
Reviewer 1 Report (Previous Reviewer 1)
There are still some issues that were not answered:
7. Please specify the calculations for the minimum sample size. I think it is important to include this number because the sheer number of participants without referring to the minimum sample does not say much.
8. Which variables were selected for stratification? How was the subsample chosen? (From section 2.1: 'For our study, 25 participants were selected using a stratified randomizing algorithm ensuring the representativeness of the data')
9. Please provide the characteristics of the population, e.g. mean age, the severity of MDD, % of different PDs.
10. Since the interview was conducted at a different time than the therapy (giving three different groups), please compare these groups. If it is discussed and accounted for under Section 2.1., please indicate the sentence.
13. The discussion needs to be strengthened, especially about other references to psychometrics. I think it is important because the study has a psychometric dimension.
Author Response
Dear reviewer
We would like to thank you again for taking the time to help us prepare the very best possible version of this manuscript. Your comments and insights have been very helpful. Please find our comments below pertaining to your specific remarks.
Please specify the calculations for the minimum sample size. I think it is important to include this number because the sheer number of participants without referring to the minimum sample does not say much.
We agree that this important detail should not have been left out in the first place. We have now added language explicitly addressing the minimum sample size, with references.
- Which variables were selected for stratification? How was the subsample chosen? (From section 2.1: 'For our study, 25 participants were selected using a stratified randomizing algorithm ensuring the representativeness of the data')
A sentence has been added explicitly outlining the procedure used.
- Please provide the characteristics of the population, e.g. mean age, the severity of MDD, % of different PDs.
Section 2.1 has been expanded to include these data.
- Since the interview was conducted at a different time than the therapy (giving three different groups), please compare these groups. If it is discussed and accounted for under Section 2.1., please indicate the sentence.
This is now explicitly discussed under section 2.1.
- The discussion needs to be strengthened, especially about other references to psychometrics. I think it is important because the study has a psychometric dimension.
New language has been added explicitly emphasizing the importance of the psychometric dimension. You will find that it is relatively brief, as we are concerned that a more lengthy exposition on this topic might dilute what we consider to be the primary focus of the article.

This manuscript is a resubmission of an earlier submission. The following is a list of the peer review reports and author responses from that submission.
Round 1
Reviewer 1 Report
1. There are different approaches (and controversy) to diagnosing personality disorders among adolescents. Please extend this topic in the introduction.
2. What is the authors' definition of an adolescent?
3. The study has a psychometric dimension. Please provide more background on psychometrics in the introduction.
4. 'Objective'- please specify what psychometric properties the authors want to investigate.
5. Why did the authors not examine validity?
6. Why did the authors decide to measure only inter-rater reliability?
7. Please specify the calculations for the minimum sample size.
8. Which variables were selected for stratification? How was the subsample chosen?
9. Please provide the characteristics of the population.
10. Since the interview was conducted at a different time than the therapy (giving three different groups), please compare these groups.
11. Please describe whether the interviewers did the joint training sessions, worked together in one centre, what are their differences, etc.?
12. Table 3 provides only 99% confidence intervals. The remaining data are repeated from Table 1.
13. The discussion needs to be strengthened, especially about other references to psychometrics.
Reviewer 2 Report
Thank you for this important contribution to our understanding of assessing adolescent mental health. Continued research following this finding will be interesting and potentially life-changing for teens and their families.
One recommendation is regarding the difference between personality disorder and personality pathology. Definitions are offered in your introduction, then mentioned in the results and discussion. However, it would be helpful to expand on these concepts in the discussion, possibly offering examples of clinical implications for how this assessment among adolescents could be transformative to both the diagnosis and pathology concepts.